# REWARD CENTERING

## ABSTRACT

We show that discounted methods for solving continuing reinforcement learning problems can be significantly improved if they center their rewards by subtracting out the rewards' (changing) empirical average. The improvement is substantial at commonly-used discount factors and increases further as the discount factor approaches 1. In addition, we show that if a *problem's* rewards are shifted by a constant, then non-centering methods perform much worse, whereas centering methods are (unsurprisingly) unaffected. In this sense, reward centering significantly increases the generality of discounted reinforcement learning methods. Insight into the benefits of reward centering can be gained from the decomposition of the discounted value function proposed by Blackwell in 1962. The reward-centering idea is general and can be added to any reinforcement learning algorithm; we showcase the centered version of the Q-learning algorithm in this paper.

## 1 DISCOUNTING: MOTIVATION AND ISSUES

There are practical difficulties in applying standard RL methods when the discount factor is close to one. These difficulties are common because in many problems of interest, we have two expectations from our solution methods: (1) learn to behave in way that maximizes rewards over the long term, (2) learn such behavior fast. These expectations appear to be conflicting when using standard methods: the rate of learning is typically high with a small discount factor but the corresponding behavior is likely too short-sighted; with large discount factors the behavior is more long-sighted but can take prohibitively long to learn. Figure 1(left) illustrates this tension on a small problem that is reflective of several real-world applications. The performance of the Q-learning algorithm (Watkins & Dayan, 1992) improved till the discount factor $\gamma = 0.9$ beyond which the rate of learning deteriorated significantly (such observations are common, for instance, see Zhang & Ross (2021)).

However, there need not be a trade-off in meeting those expectations. The right panel of Figure 1 demonstrates the performance of a variant of Q-learning which learned behavior that accrued more reward over the long term, quickly. We observed similar results on larger problems (see Section 3). There is a simple idea at play: ensure the rewards experienced by the agent are zero on average by estimating and subtracting the empirical average of the rewards online and incrementally. When this centering technique is applied to Q-learning, we call the resulting algorithm *Centered Q-learning*.

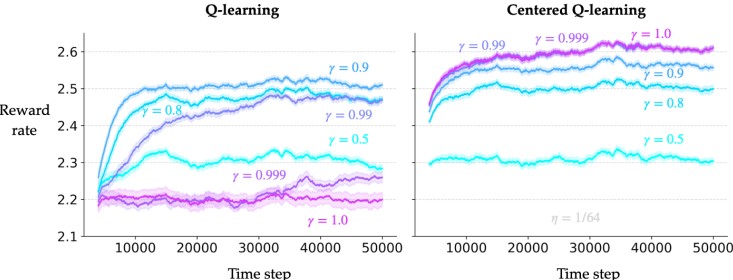

Figure 1: Learning curves corresponding to a range of discount factors for Q-learning and Centered Q-learning on the Access-Control Queuing problem. The $x$-axis denotes the number of agent-environment interactions and the $y$-axis denotes the rate of reward obtained by the agent over a moving window. The shaded region denotes one standard error. More details in-text.

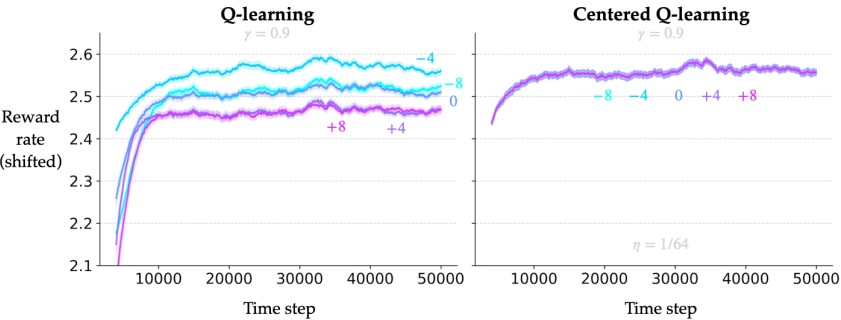

Figure 2: Learning curves for the two algorithms on slight variants of the Access-Control Queuing problem with all the rewards shifted by a constant integer. The $y$-axis is shifted to compare learning curves for all the variants on the same scale. More details in-text.

We also observed the rate of learning of standard algorithms is significantly affected by a constant shift in the rewards. Adding a constant to all the rewards does not change the ordering of the policies evaluated by the total-reward or the average-reward criterion in problems where the agent-environment interactions go on ad infinitum. However, Q-learning is susceptible to this change. To the original problem with a given reward signal, we added a constant integer to all the rewards. Five different integers result in five problem variants which have the same best policy, but Figure 2 illustrates that Q-learning behaves differently on each variant. This is undesirable, especially in the context of lifelong learning agents that may experience different kinds of reward signals over their lifetimes. On the other hand, we saw Centered Q-learning was virtually unaffected by any shift in the rewards (the learning curves for all the variants are overlapping in the right panel of Figure 2).

It turns out both of these issues are related. Asking why they occur also reveals how reward-centering helps mitigate them. So let us take a step back and ask, what is discounting and why is it desirable?

First, we formalize the interaction between the agent and the environment by a finite Markov decision process (MDP) $\mathcal{M} \doteq (\mathcal{S}, \mathcal{A}, \mathcal{R}, p)$, where $\mathcal{S}$ denotes the finite set of states, $\mathcal{A}$ denotes the finite set of actions, $\mathcal{R}$ denotes the finite set of rewards, and $p : \mathcal{S} \times \mathcal{R} \times \mathcal{S} \times \mathcal{A} \to [0, 1]$ denotes the transition dynamics. At time step $t$, the agent is in state $S_t \in \mathcal{S}$, takes action $A_t \in \mathcal{A}$ using a behavior policy $b : \mathcal{A} \times \mathcal{S} \to [0, 1]$, observes the next state $S_{t+1} \in \mathcal{S}$ and reward $R_{t+1} \in \mathcal{R}$ according to the transition dynamics $p(s', r|s, a) = \Pr(S_{t+1} = s', R_t = r|S_t = s, A_t = a)$. We consider the case where the agent-environment interaction goes on ad infinitum, as in *continuing* problems.

The discounted-reward formulation in RL is pertinent in continuing problems. The geometric discounting of future rewards is a way to ensure that the sum of infinite rewards observed by the agent starting from any state remains finite; the *total* sum of rewards for any policy in a continuing problem is infinite. Formally, for all discount factors $\gamma \in [0, 1)$, the discounted value function is finite:

$$v_\pi^\gamma(s) \doteq \mathbb{E}\left[\sum_{t=0}^\infty \gamma^t R_{t+1}|S_t = s, A_{t:\infty} \sim \pi\right] \leq R_{\max}/(1 - \gamma) \qquad \forall s, \qquad (1)$$

where $\pi : \mathcal{A} \times \mathcal{S} \to [0, 1]$ denotes a stationary policy and $R_{\max}$ denotes the upper bound of the rewards in $\mathcal{R}$. The optimal policy under the discounted-reward criterion is one that maximizes the *discounted* sum of rewards from all states. That is, for discount factor $\gamma$, $\pi_\gamma^*$ is an optimal policy if $v_{\pi_\gamma^*}^\gamma(s) \geq v_\pi^\gamma(s), \forall s \in \mathcal{S}, \forall \pi \in \Pi$, where $\Pi$ denotes the set of all stationary policies.[1]

In many problems of interest, we wish to learn behaviors that maximize the total *undiscounted* reward. But an optimal policy for a discount factor $\gamma$—call it a $\gamma$-optimal policy—does not necessarily maximize the total undiscounted reward from each state. Thankfully, an implication of Blackwell's (1962) work is that for all finite MDPs, there exists a critical $\gamma^*$ such that $\forall \gamma \in [\gamma^*, 1)$, the $\gamma$-optimal policy also maximizes the *average reward* over a long time (and hence maximizes the total

---

[1]Naik et al. (2019) and Sutton & Barto's (2018) Section 10.4 show the notion of optimality is not well-defined for the control problem in the continuing setting with function approximation. Our theoretical analysis is restricted to the tabular case for which there is a well-defined discounted-reward objective. However, we keep this caveat in mind when reporting results of discounted solution methods with function approximation later in this paper.

undiscounted reward) (see Puterman's (1994) Theorem 10.1.4 and Grand-Clément & Petrik's (2023) Theorem 4.6). However, $\gamma^*$ is unknown. So a natural choice is to use large discount factors which are hopefully larger than the unknown $\gamma^*$ for a given problem. But the rate of learning can be quite low with large discount factors. This is not just an empirical observation; it is backed by theory: the best known bounds of convergence rate for Q-learning are polynomials of $1/(1 - \gamma)$ (Qu & Wierman, 2020; Wainwright, 2019; Even-Dar et al., 2003). That is, the number of samples to learn the optimal value function with Q-learning approaches infinity as the discount factor approaches one.

The range of the discounted valued function offers a clue to why the information-theoretic bound is so large. From (1) we know the range of the discounted values is $[-R_{\max}/(1 - \gamma), R_{\max}/(1 - \gamma)]$, which grows as $\gamma$ approaches 1. However, the optimal policy is only a function of the relative (action) values, not their absolute values. So is it possible to estimate just the relative values, and would that be easier than estimating the absolute values?

The answer is affirmative thanks to the Laurent series decomposition which reveals the discounted value function comprises of a large state-independent term that does not contribute to the action selection (see Blackwell's (1962) Theorem 4a or Puterman's (1994) Corollary 8.2.4):

$$v_\pi^\gamma(s) = \frac{r(\pi)}{1 - \gamma} + \bar{v}_\pi(s) + e_\pi^\gamma(s), \qquad \forall s, \tag{2}$$

where $r(\pi)$ is the *reward rate* or the average reward obtained by policy $\pi$, $\bar{v}_\pi(s)$ denotes the *differential* value function at state $s$, each defined for ergodic MDPs (for ease of exposition) as:

$$r(\pi) \doteq \lim_{n \to \infty} \frac{1}{n} \sum_{t=1}^{n} \mathbb{E}\big[R_t \mid S_0, A_{0:t-1} \sim \pi\big], \tag{3}$$

$$\bar{v}_\pi(s) \doteq \mathbb{E}\left[\sum_{k=1}^{\infty} \big(R_{t+k} - r(\pi)\big) \mid S_t = s, A_{t:\infty} \sim \pi\right], \tag{4}$$

and $e_\pi^\gamma(s)$ denotes an error term that go to zero as the discount factor goes to one. Note the differences in the values of states do not depend on the constant term $r(\pi)/(1 - \gamma)$.

The Laurent series decomposition provides a clear insight into the issues discussed at the beginning of this section. If the state-independent *offset* has a large magnitude, approximation errors in estimating it could potentially mask the differences between the states (especially so when the value function approximators are initialized close to zero). There are two contributors to the magnitude of the offset: a large discount factor, and a large reward rate. So it is unsurprising that the practical issues discussed at the beginning of this section can be quite common, since there is a strong motivation to set large discount factors and we want to learn policies that maximize the reward rate (and hence the total reward). Large discount factors and shifts in the problems' rewards can inflate the offset, making it hard to learn the values and hence good policies.

## 2 CENTERED DISCOUNTED METHODS

If the offset $r(\pi)/(1 - \gamma)$ were zero, the value function approximator would not have to estimate it. We can arrange for that to happen. For intuition, consider the case when the agent magically knows the average reward of the target policy before we run any learning algorithm. Then, when estimating the values, if the agent subtracts this known average reward from the stream of occurring rewards, the offset would be zero, hence not contributing to the value function at all.

In particular, we can define a new *centered discounted value function* of policy $\pi$ for a state $s$ for a discount factor $\gamma \in [0, 1]$, where all the rewards are *centered* around $r(\pi)$:

$$\bar{v}_\pi^\gamma(s) \doteq \mathbb{E}\left[\sum_{t=0}^{\infty} \gamma^t \big(R_{t+1} - r(\pi)\big) \mid S_t = s, A_{t:\infty} \sim \pi\right]. \tag{5}$$

Note that as $\gamma \to 1, \bar{v}_\pi^\gamma(s) \to \bar{v}_\pi(s), \forall s$. The centered discounted value function satisfies recursive Bellman-like equations:

$$\bar{v}^\gamma(s) = \sum_a \pi(a|s) \sum_{s',r} p(s', r|s, a)\big(r - \bar{r} + \gamma \bar{v}^\gamma(s')\big), \quad \forall s, \quad \text{or} \quad \bar{\mathbf{v}}^\gamma = \mathbf{r}_\pi - \bar{r}\mathbf{1} + \gamma \mathbf{P}_\pi \bar{\mathbf{v}}^\gamma,$$

$$\tag{6}$$

where $\mathbf{1}$ denotes a vector of all ones, $\mathbf{r}_\pi$ denotes the vector of expected one-step rewards from each state, and $\mathbf{P}_\pi$ denotes policy-conditioned state-to-state transition matrix, and $(\bar{\mathbf{v}}^\gamma, \bar{r})$ are free variables. Interestingly, these Bellman equations have infinite solutions: $(\bar{\mathbf{v}}_\pi^\gamma, r(\pi))$ is a solution tuple, and so are $(\bar{\mathbf{v}}_\pi^\gamma + c\mathbf{1}, \bar{r} - c(1-\gamma))\forall c \in \mathbb{R}$.

Now if the agent wanted to estimate the centered discounted value function of a target policy $\pi$ from its stream of experience following $\pi$ and somehow knew that policy's reward rate $r(\pi)$, it could update the tabular estimates $V_t : \mathcal{S} \to \mathbb{R}$ using a variant of the TD-learning update (Sutton, 1988):

$$V_{t+1}(S_t) \doteq V_t(S_t) + \alpha\big(R_{t+1} - r(\pi) + \gamma V_t(S_{t+1}) - V_t(S_t)\big), \tag{7}$$

where $\alpha$ is the step size. However, $r(\pi)$ is typically unknown, so the agent has to estimate it as well. The most obvious way to do this is to maintain a running estimate of the reward rate at every time step. In particular, a scalar reward-rate estimate $\bar{R}$ can be updated at each time step as:

$$\bar{R}_{t+1} \doteq \bar{R}_t + \beta(R_{t+1} - \bar{R}_t), \tag{8}$$

where $\beta$ is another step size. Such an update is guaranteed to converge to true reward rate of the policy with which the agent is behaving—in this case, $\pi$. Interleaving this update with (7) would result in a convergent *on-policy* centered TD-learning algorithm (under the usual technical conditions). This is good progress, but this approach of estimating the reward rate by an exponential recency-weighted sample average of observed rewards is limited to the on-policy setting. In general the agent may behave according to a different (more exploratory) policy than the one whose values it is trying to learn. This *off-policy* setting is important to the goals of artificial intelligence (see, e.g., Sutton et al., 2011), and we are motivated to find a way to estimate the reward rate of the target policy when acting according to a different behavior policy.

We can leverage a technique from the average-reward literature. Recently, Wan et al. (2021) figured out how to estimate the reward rate in the off-policy setting for solution methods of the average-reward formulation. They showed that using the *temporal-difference* (TD) error (instead of the *conventional* error in (8)) leads to an unbiased estimate of the reward rate even in the off-policy setting. We can use this technique alongside any standard discounted method; in particular, we modify Q-learning to obtain the *Centered Q-learning* algorithm, which updates a table of action-value estimates as:

$$Q_{t+1}(S_t, A_t) \doteq Q_t(S_t, A_t) + \alpha\delta_t, \tag{9}$$

$$\bar{R}_{t+1} \doteq \bar{R}_t + \eta\alpha\delta_t, \tag{10}$$

$$\text{where,} \quad \delta_t \doteq R_{t+1} - \bar{R}_t + \gamma \max_{a'} Q_t(S_{t+1}, a') - Q_t(S_t, A_t), \tag{11}$$

$\bar{R}_t$ is the estimate of the average reward at time step $t$, and $\eta > 0$ is a step-size parameter.

Before presenting the convergence result of this algorithm, we note our reward-centering approach to minimize the effect of the state-independent offset builds on two important works. Devraj and Meyn (2021) proposed a variant of Q-learning that shifts its rewards using an arbitrary function of the estimated state-action values. In particular, the update at time $t$ is:

$$Q_{t+1}(S_t, A_t) \doteq Q_t(S_t, A_t) + \alpha\big(R_{t+1} - f(Q_t) + \gamma \max_{a'} Q_t(S_{t+1}, a') - Q_t(S_t, A_t)\big), \tag{12}$$

where, $f(Q_t) \doteq \kappa \sum_{s,a} \mu(s,a)Q_t(s,a)$, $\kappa > 0$ is a scalar, and $\mu : \mathcal{S} \times \mathcal{A} \to [0, 1]$ is a probability mass function. Let us consider the convergence of such an algorithm. For intuition, assume the shifting quantity is not changing with time like $f(Q_t)$, but is a fixed scalar $j \in \mathbb{R}$. For the MDP $\mathcal{M}$ with rewards $\mathcal{R}$, the optimal value function is $\mathbf{q}_*^\gamma$. If $j$ is subtracted from all the rewards, the agent experiences a slightly different MDP $\mathcal{M}'$ with rewards $\mathcal{R}'$, where the optimal value function is $\mathbf{q}_*^\gamma - j/(1-\gamma)\mathbf{1}$. Devraj and Meyn show their variant of Q-learning (12) converges almost surely under general conditions to $\mathbf{q}_*^\gamma - k/(1-\gamma)\mathbf{1}$, where $k$ depends on $\kappa, \mu$ and $\mathbf{q}_*^\gamma$. This is a great start. The discounted value function $\mathbf{q}_*^\gamma$ has an offset of $r(\pi_\gamma^*)/(1-\gamma)$, where $\pi_\gamma^*$ is a discounted optimal policy, and Devraj and Meyn's algorithm can remove $k/(1-\gamma)$ of it. Devraj and Meyn left the choice of $\mu$ and $\kappa$ as open questions; we show that Centered Q-learning can be seen as an instance of their algorithm with particular choices of $\mu$ and $\kappa$ that significantly reduce the effect of state-independent offset. This equivalence also enables us to use their theoretical machinery to show almost-sure convergence and strong variance-reduction properties.

Separately, Schneckenreither (2020) realized the Laurent series decomposition (2) suggests that an explicit estimate of the average reward can completely remove the offset. So they proposed an algorithm where (a) there are two discount factors to aim for the stronger Blackwell optimality, and (b) the average-reward estimate is updated like in (10) but only after non-exploratory actions. There is no convergence proof for this algorithm, though Schneckenreither analyzed that if the algorithm converged to the ideal fixed point, then the resulting policy would be (Blackwell-)optimal. Wan et al. (2021) pointed out the average-reward estimate can be updated at every time step, including ones with exploratory actions, and showed almost-sure convergence of their algorithms. Combining their insights with those from Devraj and Meyn (2021) and Schneckenreither (2020), we can show convergence of tabular Centered Q-learning. Here we present the informal theorem statement and a proof sketch; the full presentation is in Appendix B, where we also show the learned value function does not have a factor that scales with $1/(1-\gamma)$.

**Theorem 1.** *If the Markov chain induced by the stationary behavior policy is irreducible, and a per-state-action step size is reduced appropriately, Centered Q-learning (9–10) converges almost surely: $\bar{R}_t$ and $Q_t$ converge to a particular solution $(\bar{r}, \bar{q}^\gamma)$ of the following Bellman equations:*

$$\bar{q}^\gamma(s, a) = \sum_{s',r} p(s', r|s, a)\big(r - \bar{r} + \gamma \max_{a'} \bar{q}^\gamma(s', a')\big). \tag{13}$$

*Proof.* (Sketch) The proof first shows the updates of Centered Q-learning (9-10) can be combined such that it becomes an instance of Devraj and Meyn (2021)'s algorithm with particular choices of $\mu$ and $\kappa$. The convergence of the estimates then follows. Finally we show the convergent point is a particular solution among the family of solutions of (13). □

Note that the general reward-centering idea can be added to any RL algorithm for solving continuing tasks. We focus on the reward-centered versions of Q-learning in this paper; we expect similar trends for other algorithms that estimate values, like Sarsa (Rummery & Niranjan, 1994) or the family of actor-critic methods (Konda & Tsitsiklis, 2001; Schulman et al., 2016).

## 3 EMPIRICAL RESULTS

In this section, we present empirical results with both the standard Q-learning algorithm and Centered Q-learning on a set of domains with tabular, linear, and non-linear function approximation to empirically assess the benefits of reward centering. The domains are largely from CSuite: github.com/google-deepmind/csuite. We provide high-level descriptions here; the repository documentation fleshes out the details.

We begin with the Access-Control Queuing domain (Sutton & Barto, 2018). This is a continuing problem where the agent controls a server queue. A job arrives at the front of the queue with one of four priorities, and the agent has to decide at each time step whether to accept or reject the job based on the number of free servers left (out of 10). If a job is accepted, the agent gets a positive reward proportional to the job's priority; if rejected, the job is removed from the queue and the agent gets zero reward. Occupied servers get free with a certain probability at each time step, and new jobs have a uniform-random priority among $\{1, 2, 4, 8\}$. At each time step, the agent can observe the number of servers that are currently free and the priority of the job at the front of the queue.

We applied both the standard discounted Q-learning and the proposed Centered Q-learning algorithms on this domain, each for 50 independent runs of 80,000 steps. We tested various discount factors and step sizes for both algorithms, along with different values of the additional step-size parameter $\eta$ for Centered Q-learning. All the learnable parameters were initialized to zero and both algorithms used an $\epsilon$-greedy behavior policy with a fixed value of $\epsilon = 0.1$.

Figure 1 shows the online performance for both algorithms (our experiments did not have a separate testing period). For Q-learning, they correspond to the step-size parameters that resulted in the fastest learning over the training period (quantified by the area under the learning curve); for Centered Q-learning they correspond to the best step-size parameters for a fixed value of $\eta$ (shown in grey in the figure). This does not always mean the best $(\alpha, \eta)$ pair for Centered Q-learning but that is okay since the results were robust to the choice of $\eta$. Throughout the paper we follow this same practice of picking hyperparameters to plot learning curves.

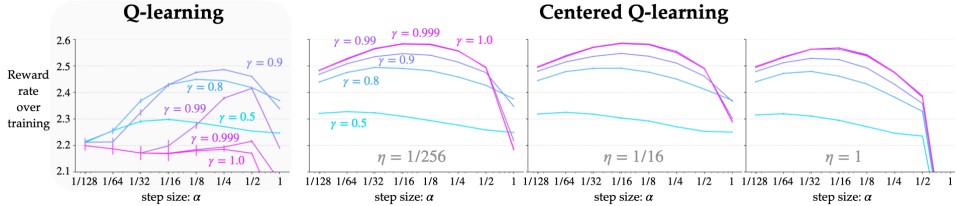

Figure 3: Parameter studies showing the sensitivity of the two algorithms' performance to their parameters on the Access-Control domain. The error bars indicate one standard error, which at times is less than the width of the lines. *Far left:* Q-learning's rate of learning deteriorated with large discount factors for a broad range of the step-size parameter $\alpha$. *Center to right:* For each discount factor, the performance of Centered Q-learning was better across a broad range of $\alpha$. Moreover, the performance was robust across a large range of its second parameter $\eta$.

We saw that the performance of Centered Q-learning did not degrade when the discount factor was close to one, unlike that with Q-learning. For each discount factor, the performance of Centered Q-learning matched or exceeded that of Q-learning. To verify if centering indeed helped remove the potentially large state-independent term, we checked the magnitude of the learned values. One way is to compute the average value across all state-action pairs. However, this approach would typically lead to a poor approximation of the magnitude of learned values because many states (especially ones with low true values) may not occur frequently in the agent's $\epsilon$-greedy interactions with the environment and hence their estimated values may stay close to their initialization. Instead, we checked the values of the states that actually occur in the agent's stream of experience, in particular the maximum action value (used to choose the argmax action) of the last $10\%$ states that occurred during training. Table 1 shows these values for the parameters that resulted in Figure 1's learning curves. As $\gamma$ increased, the magnitude of learned values increased sharply with Q-learning but remained small with Centered Q-learning.

Table 1: Magnitude of learned values by the two algorithms on Access-Control Queuing

| $\gamma$ | DiscQ | CDiscQ |
|---|---|---|
| 0.5 | 4.78 | 0.17 |
| 0.8 | 12.95 | 0.17 |
| 0.9 | 26.57 | 0.12 |
| 0.99 | 267.91 | 0.42 |
| 0.999 | 1434.47 | 0.51 |

These trends are quite general across the range of parameter values tested. Figure 3 shows the performance sensitivity to the methods' parameters. In particular, the $x$-axis denotes the step-size parameter $\alpha$ and the $y$-axis denotes the average reward obtained during the entire training period (which reflects the rate of learning). For both methods, the different curves correspond to different discount factors. For Centered Q-learning, the three plots on the right correspond to different values of the step-size parameter $\eta$. We saw the performance of Q-learning deteriorated with large discount factors for a broad range of the step-size parameter $\alpha$. In contrast, the performance of Centered Q-learning did not degrade; in fact, it improved all the way till $\gamma = 1$ for a wide range of $\eta$ values. In addition, its performance was not sensitive to the choice of $\eta$.

Figure 2 shows the behaviors of Q-learning and Centered Q-learning when applied to five problem variants with one of $\{-8, -4, 0, 4, 8\}$ added to all the rewards, which does not change the ordering of the policies in a continuing problem. Centered Q-learning, by design, behaved similarly on all the problem variants while the behavior of Q-learning was substantially different. These trends were also consistent across values of $\alpha$ and $\eta$ (see the sensitivity plots in Appendix C).

We found the reward-centering technique had similar benefits with both linear and non-linear function approximation as well (see Appendix A for all the pseudocodes).

In PuckWorld, the agent controls a puck-like object in a square rink where goal positions occur randomly. The agent can push the puck in any of the cardinal directions. Repeated actions in a direction gives the puck some velocity that is upper-bounded due to friction. The agent observes six real numbers at each time step—the puck's position and velocity and the goal position in $x$ and $y$ directions—and gets a reward proportional to the negative distance to the goal. The best policy goes to the goal position as soon as possible which moves to a new random location every 300 time steps.

We trained linear function approximators on this problem by tile-coding the 6-dimensional observation vector with 32 tilings of 4 tiles in each dimension. Each experiment was repeated for 20 runs of 300,000 steps each. We tested a range of step-size parameters and discount factors for both al-

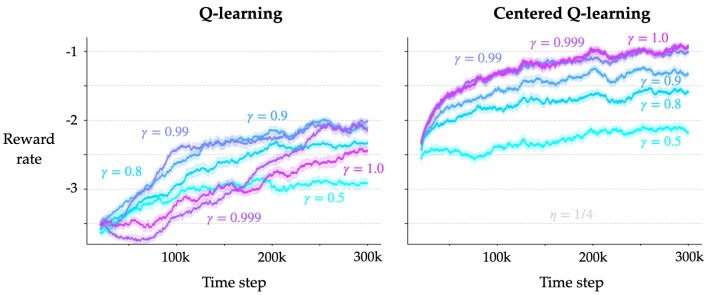

Figure 4: Learning curves corresponding to a range of discount factors for Q-learning and Centered Q-learning on the PuckWorld problem. Centered Q-learning's rate of learning was higher for each discount factor and did not degrade as $\gamma \to 1$.

gorithms, which started from zero initializations of the weights and the scalar reward-rate estimate. The behavior policy was $\epsilon$-greedy with $\epsilon = 0.1$.

The trends were similar to but more dramatic than the previous tabular experiment. Figure 4 shows learning curves corresponding to the step-size parameter that resulted in the best performance for Q-learning and the best step-size parameter for the same value of $\eta$ for Centered Q-learning (the trends were consistent across the values of $\eta$ tested; see the sensitivity plots in Appendix C). The $x$-axis denotes the number of agent-environment interactions while the $y$-axis denotes the rate of reward obtained by the agent over a moving window. The performance of Q-learning suffered as $\gamma$ increased while that of Centered Q-learning did not. In fact, it learned a better policy in shorter time for discount factors all the way to 1. The higher starting points of the learning curves also indicate that Centered Q-learning led to faster rate of learning with each of the discount factors tested.

Additionally, we observed similar trends when the rewards in the problem were shifted by a constant: Q-learning's rate of learning was highly sensitive to the shift, whereas that of Centered Q-learning was virtually unaffected, that too for a large range of its two parameters (the learning curves and sensitivity plots are in Appendix C).

The results with non-linear function approximation on the Pendulum domain followed the same trends when we tested DQN (Mnih et al., 2015) and what we call *Centered DQN*. The agent controls the torque at the base of a one-link pendulum and gets a reward at each time step proportional to the negative angular distance of the pendulum from the upright position. The pendulum starts at rest pointing down. The agent can only apply a discrete amount of torque of $\{-1, 0, 1\}$ unit at each time step after observing three real numbers: the sine and cosine of the pendulum's angle w.r.t. pointing downwards, and the pendulum's angular velocity. There are no resets or timeouts; the agent must learn to keep the pendulum in the upright position. The pendulum repeatedly falls because the upright position is an unstable equilibrium and any exploratory actions can upset the pendulum.

We tested both DQN and Centered DQN to estimate the action values in this problem. The artificial neural networks had two hidden layers with 64 units each with tanh activation functions, with the networks' weights trained using the Adam optimizer (Kingma & Ba, 2015) and the semi-gradient mean-squared-error loss. The weights were initialized in the standard way to small values around zero and the reward-rate estimate was initialized to zero. The agents followed an $\epsilon$-greedy behavior

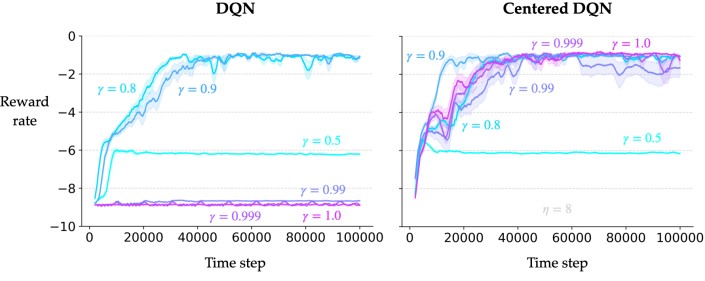

Figure 5: Learning curves corresponding to a range of discount factors for DQN and Centered DQN on the Pendulum problem. Notably, the rate of learning with Centered DQN did not degrade when $\gamma \to 1$. More details in-text.

policy with $\epsilon = 0.1$ without annealing. Each experiment was run for 100,000 steps and repeated 15 times. See Appendix C for the rest of the implementation details.

Figure 5 shows that the performance of DQN and Centered DQN with varying discount factors. A discount factor of 0.5 was too small to solve the problem. The agents learnt a good policy using DQN with discount factors 0.8 or 0.9 but failed to learn anything meaningful in 100k steps for discount factors 0.99 and larger. In contrast, with Centered DQN the rate of learning did not degrade even with discount factors all the way up to 1.

Here too we observed similar trends when the two algorithms were tested on reward-shifted variants of the Pendulum problem: DQN's rate of learning was highly sensitive to the shift, whereas that of Centered DQN was robust across a broad range of its parameters (see Appendix C).

We also performed a series of experiments with the Catch domain, using both linear and non-linear function approximators. In Catch, the agent controls a crate at the bottom row of a 2D pixel grid to catch falling fruits. The agent gets a +1 reward on successfully catching a fruit, -1 on dropping one, and 0 otherwise. At each time step, a new fruit is spawned with 10% probability in the top row, in a random column. More than one fruit may be falling at any point of time, and each fruit falls one pixel in one time step. The agent can choose among three actions: move the crate one pixel right, left, or stay put. There are two kinds of observation vectors available to an agent: a 3-dimensional real vector containing the $x$ coordinate of the crate and the $(x, y)$ coordinates of the lowermost fruit; a 50-dimensional binary vector which is the flattened version of the entire $10 \times 5$ pixel grid.

Figure 6 shows the learning curves of linear Q-learning and Centered Q-learning when applied to the Catch problem by tile-coding the 3D real-valued observation vectors. The two plots on the left show learning curves on the original problem. Both Q-learning and Centered Q-learning resulted in good policies for all the discount factors tested, including a relatively low discount factor of 0.5. But as soon as the problem's rewards were shifted, the performance of Q-learning suffered significantly for larger discount factors. The two plots on the right of Figure 6 show the performance on a variant of the Catch problem which had 2 subtracted from all the rewards. Recall that shifting the rewards by a constant does not change the ordering of the policies—the best policy remains unchanged. The rate of learning was much slower for discount factors larger than 0.9 with Q-learning for all values of the step size tested; on the other hand, Centered Q-learning continued to perform well.

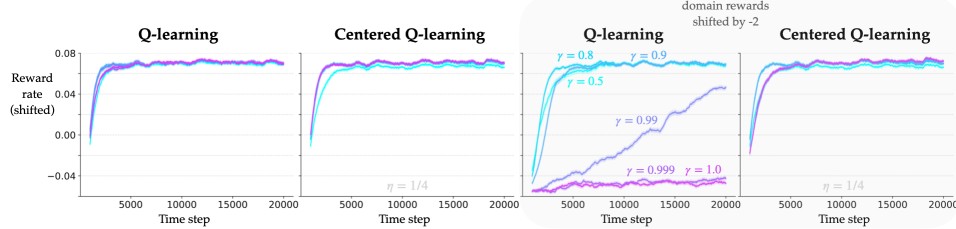

Figure 6: Learning curves of the two linear algorithms for various discount factors on two variants of Catch. *Left:* On the original problem, both algorithms performed well for all discount factors tested. *Right:* On the problem variant with all rewards shifted by -2, Q-learning's rate of learning was much lower for larger discount factors while that of Centered Q-learning was virtually unaffected.

These trends are further supplemented by the two plots in the left of Figure 7, where we show the sensitivity of the two algorithms to different problem variants. We created five problem variants by adding each of $\{-4, -2, 0, 2, 4\}$ to all the rewards one at a time. When the rewards are shifted by zero, we get the original problem; when the rewards are shifted by -2, we get the other problem variant discussed in Figure 6. On the $x$-axis is the effective step size for the linear function approximators and on the $y$-axis is the reward rate averaged over the entire training period. The $y$-axis is adjusted to compare the performance on all the problem variants at the same scale (see Appendix C for more details). The performance of Q-learning was problem-dependent, whereas Centered Q-learning resulted in roughly the same rate of learning regardless of the problem variant.

The results with non-linear function approximation also exhibited the same trends. This time the agents observed the 50D binary vector and estimated the values of the three actions with networks having a hidden layer of 128 units. Each experiment was run for 80,000 steps and repeated 15

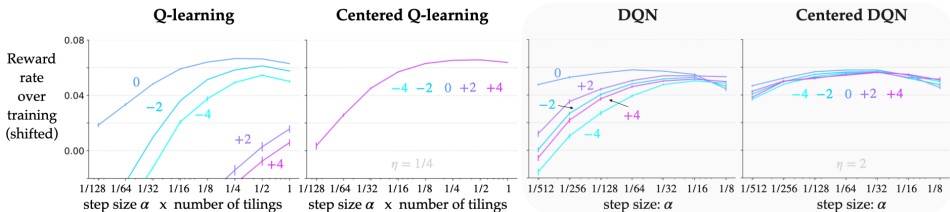

Figure 7: Parameter studies showing the sensitivity of the algorithms to their step-size parameter and to variants of the Catch problem. *Left:* Q-learning's rate of learning depended strongly on the problem variant and the step size, whereas that of Centered Q-learning did not depend on the problem variant. *Right:* The rate of learning of Centered DQN was roughly independent of the problem variant as well as the step-size parameter while that of DQN depended on both.

times, starting with the standard initialization of all the weights to small values around zero and the reward-rate estimate to zero. The remaining agent details were same as those for Pendulum.

The two plots on the right of Figure 7 show that the rate of learning with Q-learning varied widely across the range of step sizes and across the range of problem variants. In particular, the rate of learning was fastest on the original problem for a broad range of the step sizes, although the best step size for each problem variant resulted in roughly the same rate of learning. With Centered Q-learning, the rate of learning was almost independent of the problem variant, that too for a broad range of the step size $\alpha$.

The main takeaways from these results (and additional ones in Appendix C) across different problems with tabular, linear, and non-linear function approximators are:

- Reward centering can improve the performance of Q-learning-like methods for all discount factors, especially as $\gamma \to 1$.
- Reward centering can also make these methods robust to shifts in the problems' rewards.
- The parameter $\eta$ for reward centering can be relatively easy to set.

## 4 DISCUSSION AND LIMITATIONS

Centering the rewards around the current average reward makes standard methods faster and easier to use. The benefits stem from minimizing the effect of the state-independent offset that is revealed by the Laurent series decomposition of the discounted value function. One of the biggest appeals of centering is that the discount factor becomes less of a parameter that needs to be tuned: it can be set to large value (like 1) without the fear of a massive reduction in learning speed.

That being said, further improvements can be made. Reward centering helps value function approximators to focus on the essential differences between the states rather than the differences in addition to a potentially large offset. However, the relative values themselves may have large magnitude. For instance, if all the rewards in a problem are multiplied by a large number, then the relative values will be scaled by the same number. If there are problems in estimating a large range of relative values, then reward centering can be combined with clever scaling approaches. For instance, Schaul et al. (2021) have proposed a lightweight trick to rescale TD errors into an optimization-friendly range (building on prior work by van Hasselt et al. (2016) and Pohlen et al. (2018)). With some care, their techniques can likely be extended from the episodic to the continuing case.

Speaking of the episodic case, a pertinent direction of future work is to extend the general idea (and benefits) of reward centering from the continuing to the episodic case. Note that unlike in continuing problems, shifting the rewards (by reward centering or otherwise) can change the episodic problem—the ordering of policies may change. However, a stream of experience from long episodes can appear as continuing to a learning agent, so perhaps there is an equivalent technique that makes it easy to learn policies that maximize the total undiscounted in episodic problems.

The reward-centering idea can be added to any continuing RL algorithm that estimates values. Studying the theory and empirical performance of centered versions of common algorithms other than Q-learning would further showcase the benefits of reward centering. Finally, while we obtained promising empirical results when centering with function approximation, the theory of reward centering should be analyzed beyond the tabular case.

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

## A  PSEUDOCODES

---

**Algorithm 1:** Centered Q-learning (tabular)

---
**Input:** The policy $b$ to be used (e.g., $\epsilon$-greedy)
**Algorithm parameters:** discount factor $\gamma$, step-size parameters $\alpha, \eta$
1 Initialize $Q(s, a) \ \forall s, a; \bar{R}$ arbitrarily (e.g., to zero)
2 Obtain initial $S$
3 **while** *still time to train* **do**
4     $A \leftarrow$ action given by $b$ for $S$
5     Take action $A$, observe $R, S'$
6     $\delta = R - \bar{R} + \gamma \max_a Q(S', a) - Q(S, A)$
7     $Q(S, A) = Q(S, A) + \alpha\delta$
8     $\bar{R} = \bar{R} + \eta\alpha\delta$
9     $S = S'$
10 **end**

---

---

**Algorithm 2:** Centered Q-learning (linear)

---
**Input:** The policy $b$ to be used (e.g., $\epsilon$-greedy)
**Algorithm parameters:** discount factor $\gamma$, step-size parameters $\alpha, \eta$
1 Initialize $\mathbf{w}_a \in \mathbb{R}^d \ \forall a, \bar{R}$ arbitrarily (e.g., to zero)
2 Obtain initial observation $\mathbf{x}$
3 **while** *still time to train* **do**
4     $A \leftarrow$ action given by $b$ for $\mathbf{x}$
5     Take action $A$, observe $R, \mathbf{x}'$
6     $\delta = R - \bar{R} + \gamma \max_a \mathbf{w}_a^\top \mathbf{x} - \mathbf{w}_A \mathbf{x}$
7     $\mathbf{w}_A = \mathbf{w}_A + \alpha\delta\mathbf{x}$
8     $\bar{R} = \bar{R} + \eta\alpha\delta$
9     $\mathbf{x} = \mathbf{x}'$
10 **end**

---

---

**Algorithm 3:** Centered DQN (non-linear)

---
**Input:** The policy $b$ to be used (e.g., $\epsilon$-greedy)
**Algorithm parameters:** discount factor $\gamma$, step-size parameters $\alpha, \eta$
1 Initialize value network, target network; initialize $\bar{R}$ arbitrarily (e.g., to zero)
2 Obtain initial observation $\mathbf{x}$
3 **while** *still time to train* **do**
4     $A \leftarrow$ action given by $b$ for $\mathbf{x}$
5     Take action $A$, observe $R, \mathbf{x}'$
6     Store tuple $(\mathbf{x}, A, R, \mathbf{x}')$ in the experience buffer
7     **if** *time to update estimates* **then**
8        Sample a minibatch of transitions $(\mathbf{x}, A, R, \mathbf{x}')^b$
9        For every $i$-th transition: $\delta_i = R_i - \bar{R} + \gamma \max_a \hat{q}(\mathbf{x}'_i, a) - \hat{q}(\mathbf{x}_i, A_i)$
10        Perform a semi-gradient update of the value network parameters with a loss fn of $\delta^2$
11        $\bar{R} = \bar{R} + \eta\alpha \, \mathrm{mean}(\delta)$
12        Update the target network occasionally
13     **end**
14     $\mathbf{x} = \mathbf{x}'$
15 **end**

---

We recommend two small but useful optimizations to these general pseudocodes in Appendix C.

# B CONVERGENCE OF CENTERED Q-LEARNING

This section presents the complete Centered Q-learning algorithm, the convergence theorem, and its proof.

Suppose the agent's interaction with the MDP follows a stationary behavior policy $b \in \Pi$. Let $S_t, A_t$ denote the state-action pair occurring at time step $t$, followed by the reward $R_{t+1}$ and next state $S_{t+1}$. Let $\nu_t(s, a)$ denote the number of times a state-action pair $(s, a)$ has occurred up to and including time step $t$. The update rules of Centered Q-learning are:

$$Q_{t+1}(S_t, A_t) \doteq Q_t(S_t, A_t) + \alpha_{\nu_t(S_t, A_t)} \delta_t, \tag{14}$$

$$\bar{R}_{t+1} \doteq \bar{R}_t + \eta \alpha_{\nu_t(S_t, A_t)} \delta_t, \tag{15}$$

$$\text{where,} \quad \delta_t \doteq R_{t+1} - \bar{R}_t + \gamma \max_{a'} Q_t(S_{t+1}, a') - Q_t(S_t, A_t), \tag{16}$$

$\eta > 0$, and $\alpha_n = c/(n+d)$ where $c, d > 0$ for all $n \geq 1$.[2]

**Theorem 1.** (Formal) *If the joint process $\{S_t, A_t\}$ induced by the stationary behavior policy is an irreducible Markov chain, that is, starting from every state-action pair, there is a non-zero probability of transitioning to any other state-action pair in a finite number of steps, then $(Q_t, \bar{R}_t)$ in tabular Centered Discounted Q-learning (14)–(16) converges to a solution of $(\bar{\mathbf{q}}^\gamma, \bar{r})$ in (13).*

*Proof.* Recall the update equation for Devraj and Meyn's (2021) Relative Q-learning:

$$Q_{t+1}(S_t, A_t) \doteq Q_t(S_t, A_t) + \alpha \big( R_{t+1} - \kappa \boldsymbol{\mu}^\top \mathbf{Q}_t + \max_{a'} Q_t(S_{t+1}, a') - Q_t(S_t, A_t) \big). \tag{17}$$

Note from (14) and (15) that $\bar{R}_t - \bar{R}_0 = \eta \big( \sum_{s,a} Q_t(s, a) - \sum_{s,a} Q_0(s, a) \big)$. To simplify the analysis, we can assume $\bar{R}_0 = 0$ and $\mathbf{Q}_0 = \mathbf{0}$ without loss of generality. Then, $\bar{R}_t = \eta \sum_{s,a} Q_t(s, a)$. Thus, we can rewrite (14) and (15) as:

$$Q_{t+1}(S_t, A_t) \doteq Q_t(S_t, A_t) + \alpha \big( R_{t+1} - \eta \sum_{s,a} Q_t(s, a) + \max_{a'} Q_t(S_{t+1}, a') - Q_t(S_t, A_t) \big). \tag{18}$$

Comparing (17) and (18), we can write Centered Q-learning as an instance of Relative Q-learning with:

$$\mu(s, a) = \frac{1}{|\mathcal{S}||\mathcal{A}|} \, \forall s, a, \quad \text{and} \quad \kappa = \eta |\mathcal{S}||\mathcal{A}|.$$

Then the convergence result of Relative Q-learning applies. Thus,

$$Q_t \to Q_\infty \doteq \mathbf{q}_*^\gamma - \frac{\kappa}{1 + \kappa - \gamma} \boldsymbol{\mu}^\top \mathbf{q}_*^\gamma \mathbf{1}$$

$$= \mathbf{q}_*^\gamma - \frac{\eta}{1 + \eta |\mathcal{S}||\mathcal{A}| - \gamma} \sum_{s,a} q_*^\gamma(s, a) \mathbf{1}. \tag{19}$$

And,

$$\bar{R}_t \to \bar{R}_\infty \doteq \eta \sum_{s,a} q_*^\gamma(s, a) - \frac{\eta^2 |\mathcal{S}||\mathcal{A}|}{1 + \eta |\mathcal{S}||\mathcal{A}| - \gamma} \sum_{s,a} q_*^\gamma(s, a)$$

$$= \frac{\eta(1 - \gamma)}{1 + \eta |\mathcal{S}||\mathcal{A}| - \gamma} \sum_{s,a} q_*^\gamma(s, a). \tag{20}$$

Now, note that the family of solutions of (13) is $\big( \bar{\mathbf{q}}_*^\gamma + c\mathbf{1}, r(\pi_\gamma^*) - c(1 - \gamma) \big)$, or equivalently $\big( \mathbf{q}_*^\gamma - d\mathbf{1}, d(1 - \gamma) \big)$. Choosing $d = \frac{\eta}{1 + \eta |\mathcal{S}||\mathcal{A}| - \gamma} \sum_{s,a} q_*^\gamma(s, a)$, we see that (19) and (20) form a solution tuple of (13).

$\square$

---

[2]Devraj and Meyn (2021) considered the step-size sequence $1/n$ in their algorithm but it can be easily verified that $\alpha_n = c/(n+d)$ also satisfies the step-size condition required by Borkar and Meyn's (2000) seminal result (that was used by Devraj & Meyn (2021) to show the convergence of their algorithm).

**Note:** the use of the tight 'irreducible' assumption is not due to theoretical necessity but for ease of presentation. An irreducible Markov chain is one in which there is a non-zero probability of going from every state to every state (each pair of states is then said to *communicate* with each other). A looser 'weakly communicating' assumption means that apart from the closed irreducible set of states, there are some *transient* states that eventually lead to the closed set. The irreducible assumption is common in non-episodic MDP literature because convergence can then be specified for every state without separately specifying what happens to transient and non-transient states. Devraj and Meyn (2021) use the irreducible assumption (as assumption Q1 in their paper) and our result builds on their work and hence uses the same assumption. See Puterman's (1994) Appendix A.2 for a detailed classification of states of a Markov chain.

REDUCTION OF THE OFFSET DUE TO CENTERED Q-LEARNING

This subsection shows how Centered Discounted Q-learning reduces the state-independent offset term in the estimated $Q$ values.

Let $\pi_\gamma^*$ be a $\gamma$-optimal policy. Assume the Markov chain under $\pi_\gamma^*$ is unichain, meaning that there is only one recurrent class of states and a set of transient states in the chain. Let $r(\pi_\gamma^*)$ denote its reward rate. Note that $q_{\pi_\gamma^*}^\gamma = q_*^\gamma$. We can rewrite (19) as:

$Q_\infty(s,a)$

$$= q_{\pi_\gamma^*}^\gamma(s,a) - \frac{\eta}{1 + \eta|\mathcal{S}||\mathcal{A}| - \gamma} \sum_{s,a} q_{\pi_\gamma^*}^\gamma(s,a) + \frac{r(\pi_\gamma^*)}{1-\gamma} - \frac{r(\pi_\gamma^*)}{1-\gamma}$$

$$= \left( q_{\pi_\gamma^*}^\gamma(s,a) - \frac{r(\pi_\gamma^*)}{1-\gamma} \right) - \frac{\eta}{1 + \eta|\mathcal{S}||\mathcal{A}| - \gamma} \sum_{s,a} q_{\pi_\gamma^*}^\gamma(s,a) + \frac{r(\pi_\gamma^*)}{1-\gamma}$$

$$= \left( \bar{q}_{\pi_\gamma^*}(s,a) + \epsilon(\gamma,s,a) \right) - \frac{\eta}{1 + \eta|\mathcal{S}||\mathcal{A}| - \gamma} \sum_{s,a} q_{\pi_\gamma^*}^\gamma(s,a) + \frac{r(\pi_\gamma^*)}{1-\gamma}$$

$$= \bar{q}_{\pi_\gamma^*}(s,a) + \epsilon(\gamma,s,a) - \frac{\eta}{1 + \eta|\mathcal{S}||\mathcal{A}| - \gamma} \sum_{s,a} \left( q_{\pi_\gamma^*}^\gamma(s,a) - \frac{r(\pi_\gamma^*)}{1-\gamma} \right) + \frac{r(\pi_\gamma^*)}{1-\gamma} \left( 1 - \frac{\eta|\mathcal{S}||\mathcal{A}|}{1 + \eta|\mathcal{S}||\mathcal{A}| - \gamma} \right)$$

$$= \bar{q}_{\pi_\gamma^*}(s,a) + \epsilon(\gamma,s,a) - \frac{\eta}{1 + \eta|\mathcal{S}||\mathcal{A}| - \gamma} \sum_{s,a} \left( q_{\pi_\gamma^*}^\gamma(s,a) - \frac{r(\pi_\gamma^*)}{1-\gamma} \right) + \frac{r(\pi_\gamma^*)}{1-\gamma} \left( \frac{1-\gamma}{1 + \eta|\mathcal{S}||\mathcal{A}| - \gamma} \right)$$

$$= \bar{q}_{\pi_\gamma^*}(s,a) + \epsilon(\gamma,s,a) - \frac{\eta}{1 + \eta|\mathcal{S}||\mathcal{A}| - \gamma} \sum_{s,a} \left( \bar{q}_{\pi_\gamma^*}(s,a) + \epsilon(\gamma,s,a) \right) + \frac{r(\pi_\gamma^*)}{1 + \eta|\mathcal{S}||\mathcal{A}| - \gamma}$$

where the third and last equation uses the Laurent series decomposition. As $\gamma \to 1$, $\pi_\gamma^*$ becomes a Blackwell-optimal policy (denote it by $\pi_B$) and $\epsilon(\gamma,s,a) \to 0$, so the above equation converges to:

$$\bar{q}_{\pi_B}(s,a) - \frac{1}{|\mathcal{S}||\mathcal{A}|} \sum_{s,a} \bar{q}_{\pi_B}(s,a) + \frac{r(\pi_B)}{\eta|\mathcal{S}||\mathcal{A}|}.$$

The first term is the differential action-value function of $\pi_B$. The third term is negligible in problems with large state and action spaces. The second term is the mean of differential action-values. In general this can be non-zero. But we can build intuition for its magnitude by considering a special case when the stationary distribution $\mu_{\pi_B}$ (that uniquely exists under the unichain assumption) is uniform: $\mu_{\pi_B}(s,a) = 1/(|\mathcal{S}||\mathcal{A}|)$.[3] Then this second term is also zero (by Wan et al.'s (2021) Lemma B.11).

Importantly, none of the three terms scale with $1/(1-\gamma)$. Hence the magnitude of $Q_\infty$ is typically much smaller than $q_*^\gamma$, especially when $\gamma \to 1$.

---

[3]This special case also helps build intuition of where the reward-rate estimate converges: from (20), $\bar{R}_\infty = r(\pi_\gamma^*) * \eta|\mathcal{S}||\mathcal{A}|/(1 - \gamma + \eta|\mathcal{S}||\mathcal{A}|) \to r(\pi_\gamma^*)$, for problems with large state and/or action spaces. The equality is a result of the property: $\mu_\pi^\top v_\pi^\gamma = r(\pi)/(1-\gamma)$ (Singh et al., 1994; Sutton & Barto, 2018: Section 10.4).

## C    MORE RESULTS AND EXPERIMENTAL DETAILS

In this section we provide the remaining experimental details and additional results that supplement the ones in the main text.

Table 2 contains a list of all the hyperparameters tested for the hyperparameters that are common across all the domains: $\gamma, \alpha, \eta$. Note that we can obtain the Q-learning algorithm from the implementation of the Centered Q-learning algorithm if the reward-rate estimate is initialized to zero and $\eta = 0$. Also note that the non-linear centering algorithm (Centered DQN) in its current form requires a large value of $\eta$ compared to the the tabular or linear versions. The reason is how a minibatch is used in the implementation of this deep RL algorithm. In line 11 of Algorithm 3, the mean of the TD errors of the minibatch of transitions is taken. The mean can make the overall gradient for the reward-rate update very small, so a large value of $\eta$ can be used.

The number of timesteps, number of runs, initializations are reported in the main text. The agent's behavior policy was always $\epsilon$-greedy with fixed $\epsilon = 0.1$. We set commonly used values for the various parameters of the deep RL (non-linear) experiments: the batch size was 64, the value-network and reward-rate parameters were updated every 32 steps, the target network was updated every 128 steps, the experience buffer size was 10,000. Apart for the main step-size parameter, the default parameters (set by PyTorch (Paszke et al., 2019)) were used for the Adam optimizer.

In our implementations we added two simple optimizations:

1. Make the reward-rate estimate completely independent of its initialization: this can be done using the unbiased constant step-size trick (see Sutton & Barto's (2018) Exercise 2.7).

2. Propagate the changes to the reward-rate estimate faster: this can be done by first computing the TD error, then updating the reward-rate estimate, then recomputing the TD error with the new reward-rate estimate, and finally updating the value estimate(s).

These optimizations did not affect the overall trends in the results but provided a small yet noticeable improvement for a tiny computational cost, hence we recommend using them.

For the experiments involving a shift in the problem rewards, the rewards obtained on each problem variant are not directly comparable. For intuition, imagine the first four rewards in the original problem be 2,0,3,1. In a variant of the problem with 5 added to all the rewards, the first four rewards may now appear to be 7,5,8,4. An agent solving the latter problem might trivially appear better than one solving the former problem even though its fourth reward was relatively lower. To compare them meaningfully, from the rewards obtained by an agent, we can subtract the constant that was added in the first place to all the problem's rewards. That is, we can shift the rewards back to make fair comparisons across problem variants. This is what we did when presenting the results of the shifting experiments; this is explicitly denoted by the word "shifted" in the $y$-axis label.

Table 2: List of hyperparameters tested for each domain

|  | $\gamma$ | $\alpha$ | $\eta$ |
|---|---|---|---|
| **Access-Control Queuing** (tabular) | [0.5, 0.8, 0.9, 0.99, 0.999, 1] | [1/128, 1/64, 1/32, 1/16, 1/8, 1/4, 1/2, 1] | [0,1/256, 1/64, 1/16,1/4, 1] |
| **PuckWorld** (linear) | [0.5, 0.8, 0.9, 0.99, 0.999, 1] | [0.01, 0.1, 0.3, 0.5, 0.7, 0.9, 1.0, 1.1] | [0,1/256, 1/64, 1/16,1/4, 1] |
| **Catch** (linear) | [0.5, 0.8, 0.9, 0.99, 0.999, 1] | [1/128, 1/64, 1/32, 1/16, 1/8, 1/4, 1/2, 1] | [0,1/256, 1/64, 1/16,1/4, 1] |
| **Catch** (non-linear) | [0.5, 0.8, 0.9, 0.99, 0.999, 1] | [1/512, 1/256, 1/128, 1/64, 1/32, 1/16, 1/8] | [0, 1, 2, 4, 8, 16] |
| **Pendulum** (non-linear) | [0.5, 0.8, 0.9, 0.99, 0.999, 1] | [1/512, 1/256, 1/128, 1/64, 1/32, 1/16, 1/8] | [0, 1, 2, 4, 8, 16] |

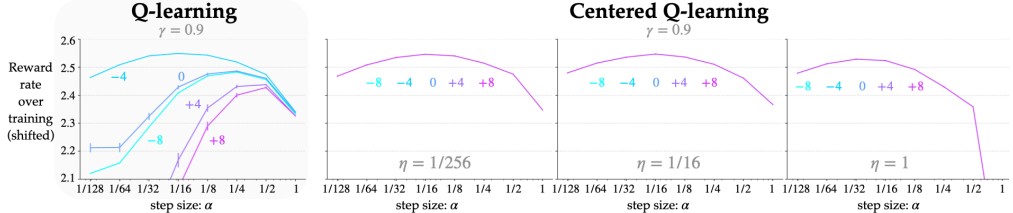

Figure 8: Parameter studies showing the sensitivity of the two algorithms' performance to variants of the Access-Control Queuing domain. The error bars indicate one standard error, which at times is less than the width of the lines. *Far left:* Q-learning's performance differed significantly on the different variants over a broad range of the step-size parameter $\alpha$. *Center to right:* Centered Q-learning performance was about the same across the problem variants, and was quite robust to the choice of its parameter $\eta$. All the curves correspond to $\gamma = 0.9$; the trends were consistent across other discount factors.

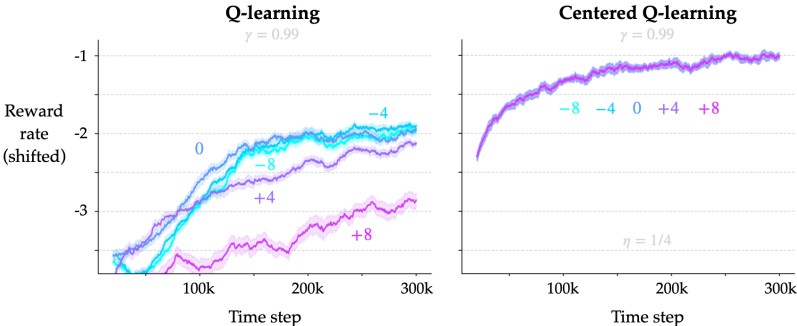

Figure 9: Learning curves for Q-learning and Centered Q-learning with $\gamma = 0.99$ on variants of the PuckWorld problem. The performance of Q-learning was different on each variant while that of Centered Q-learning was roughly the same. Centered Q-learning also resulted in much faster learning. These trends were consistent across values of $\gamma$.

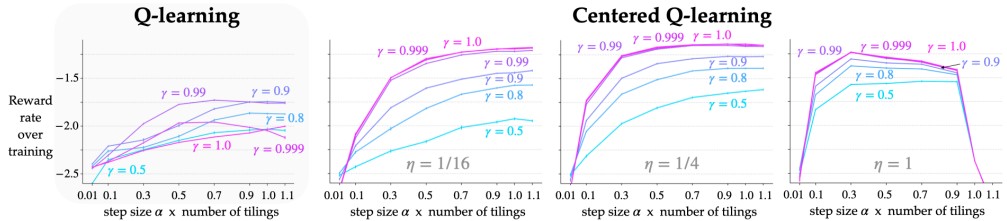

Figure 10: Parameter studies showing the sensitivity of the two algorithms' performance to their parameters on the PuckWorld domain. *Far left:* Q-learning's performance was relatively poor for a large range of $\alpha$. *Center to right:* For each discount factor, the performance of Centered Q-learning was better across a broad range of $\alpha$. Moreover, performance only changed a little w.r.t. $\eta$.

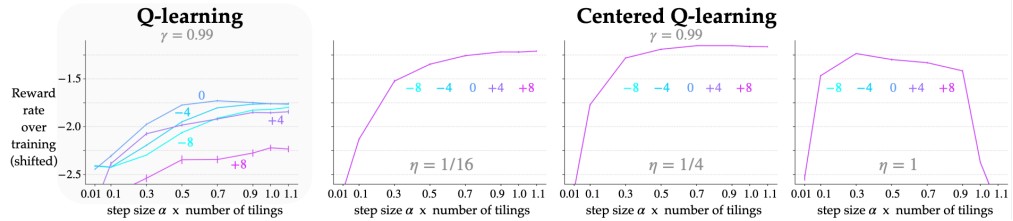

Figure 11: Parameter studies showing the sensitivity of the two algorithms' performance to variants of the PuckWorld domain. The error bars indicate one standard error, which at times is less than the width of the lines. *Far left:* Q-learning's performance differed significantly on the different variants over a broad range of the step-size parameter $\alpha$. *Center to right:* Centered Q-learning performance was about the same across the problem variants, and was quite robust to the choice of its parameter $\eta$. All the curves correspond to $\gamma = 0.99$; the trends were consistent across other discount factors.

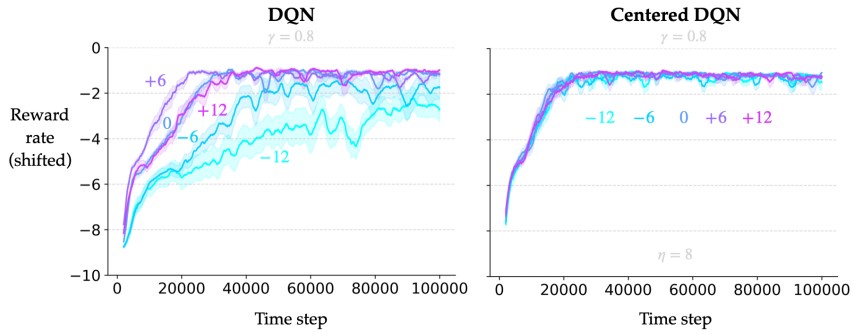

Figure 12: Learning curves for DQN and Centered DQN with $\gamma = 0.8$ on variants of the Pendulum problem. The performance of DQN was different on each variant while that of Centered DQN was roughly the same. Centered DQN also resulted in much faster learning.

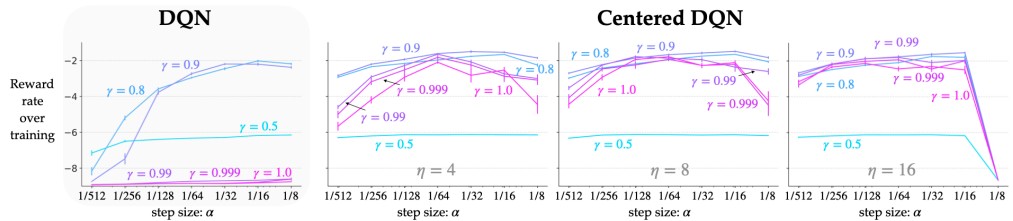

Figure 13: Parameter studies showing the sensitivity of the two algorithms' performance to their parameters on the Pendulum domain. *Far left:* The performance of DQN was not good for discount factors larger than 0.9. *Center to right:* For each discount factor, the performance of Centered Q-learning was better across a broad range of $\alpha$. And CDQN was not too sensitive to its second parameter $\eta$. $\gamma = 0.5$ was too small to solve this problem.

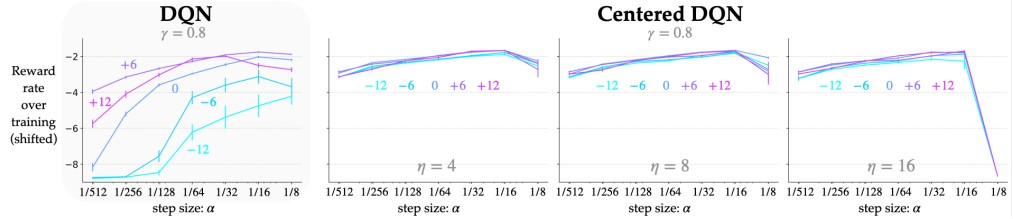

Figure 14: Parameter studies showing the sensitivity of the two algorithms' performance with $\gamma = 0.8$ to variants of the Pendulum problem. *Far left:* DQN's performance differed significantly on the different variants. *Center to right:* Centered DQN's performance was about the same across the problem variants across a large range of the step size $\alpha$, and was also quite robust to the choice of $\eta$.

# D    RELATION TO OTHER APPROACHES

The reward-centering idea is compatible with (and not a competitor of) approaches like reward scaling and advantage estimation. In addition, reward centering is a particular type of reward shaping. We discuss these points in the following subsections.

### REWARD CENTERING IS COMPATIBLE WITH REWARD SCALING

Dividing all the rewards with a (potentially changing) scalar number is typically referred to as reward scaling (see, e.g., Engstrom et al., 2020). Just like reward centering, reward scaling does not change the ordering of policies in a continuing problem. Scaling reduces the spread of the rewards, centering brings them close to zero, both of which can be favorable to complex function approximators such as artificial neural networks that are used for value estimation starting from a close-to-zero initialization. The popular stable_baselines3 repository scales (and clips[4]) the rewards by a running estimate of the variance of the discounted returns (github.com/DLR-RM/stable-baselines3/blob/master/stable_baselines3/common/vec_env/vec_normalize.py#L256). Mean-centering the rewards as well would be beneficial for continuing domains.

Note that the mechanism of computing the mean and variance is trickier in the off-policy case than the on-policy case. We use Wan et al.'s (2021) technique to estimate the mean, which is theoretically sound even in the off-policy case. Simply maintaining a running estimate of the variance (as in the stable_baselines' approach) introduces a bias. Schaul et al.'s (2021) technique is a good starting point.

### REWARD CENTERING IS COMPATIBLE WITH ADVANTAGE ESTIMATION

Reward centering and the advantage function have orthogonal benefits. The advantage function benefits the actor by reducing the variance of the updates in the policy space (Sutton & Barto, 2018; Schulman et al., 2016). On the other hand, reward centering benefits the critic's or baseline's estimation by eliminating the need to estimate the large state-independent constant offset. Both the quantities involved in the advantage function—$a_\pi^\gamma(s,a) = q_\pi^\gamma(s,a) - v_\pi^\gamma(s) \; \forall s, a$—have the large state-independent offset $r(\pi)/(1 - \gamma)$. The net effect of the offset is zero when they are subtracted. But the key point is that both the state- and action-value estimates include the large offset. Reward centering removes the need to estimate the large offset for both the state- and action-value function, which simplifies the critic-estimation problem. The actor update is left unchanged with reward centering because the advantage function itself remains unchanged: $\bar{a}_\pi^\gamma(s,a) = \bar{q}_\pi^\gamma(s,a) - \bar{v}_\pi^\gamma(s)$, because $\bar{q}_\pi^\gamma(s,a) = q_\pi^\gamma(s,a) - r(\pi)/(1 - \gamma)$ and $\bar{v}_\pi^\gamma(s) = v_\pi^\gamma(s) - r(\pi)/(1 - \gamma)$ (using this paper's notation).

Hence we expect reward centering to benefit all the algorithms that estimate values, which includes all actor-critic methods that involve advantage estimation. An extensive empirical study of centered variants of several common policy-based algorithms (like TRPO and PPO) is a ripe avenue of future work.

### REWARD CENTERING IS A SPECIFIC TYPE OF REWARD SHAPING

Reward centering can be seen as reward shaping (Ng et al., 1999) with a constant state-independent potential function: $\Phi(s) = r(\pi)/(1 - \gamma) \; \forall s$. Their Theorem 1 then reiterates that reward centering does not change the optimal policy of the problem.

A possible drawback of reward shaping is that fully specifying the potential-based shaping function can be tricky, especially for problems with large state spaces. In the case of reward centering this is relatively easy: the potential function is constant across the entire state space, and we know how to learn the average reward reliably from data.

---

[4]Reward clipping in general changes the problem. Blinding the agent from large rewards can impose a performance ceiling or make some games impossible to solve (Schaul et al.'s (2021) Section 4.3 discusses this in the context of Atari problems).

