# OpenReview forum: "Reward Centering"
_ICLR.cc/2024/Conference — Submitted to ICLR 2024_

### Official Review · Reviewer_xMY5 · 2023-10-15

**Soundness:** 3 good
**Presentation:** 1 poor
**Contribution:** 2 fair
**Rating:** 5
**Confidence:** 5

**Summary:**

This paper investigates the effect of reward centering and how it can affect learning speed and accuracy in value-based methods. Specifically, it is shown how subtracting the average reward estimate from the reward values can improve the learning speed and allow to use higher discount factors, while at the same time making the learning algorithm robust to shifts in the reward function. Empirical results show the wide applicability of this idea, both with and without function approximation.

**Strengths:**

Although not a major breakthrough, the paper investigates an interesting aspect that can improve the learning performance of many RL algorithms. Indeed, it is interesting to note how simply shifting the reward function by a certain value can completely hinder the performance of the Q-learning algorithm, even though this does not change the policy ordering at all. In this sense, the empirical results are really useful at showing the merits of the proposed idea, as well as its applicability. Also, the explanation of this idea is quite clear and easy to follow.

**Weaknesses:**

However, the structure of the paper is not really good to drive a reader through the proposed concepts and idea. A more organic and clear form should be adopted to make the work much more understandable and readable. Please see the Questions below for a broader discussion on this.

**Questions:**

- Having the paper to start with a (simple) experiment demonstrating the effectiveness of the proposed idea, without either introducing it or giving any explanation before, is weird: it is difficult to contextualize its significance or assess any proper merit here. I would recommend to start the paper with a proper Introduction section to present the investigated setting and motivations for that, to then show and analyse results after these have been already clarified to the reader.
- In general, the paper is not well structured. It starts by diving directly into the investigated problem, without giving any context to that, not presenting any Background section to inform the reader about useful notions and concepts, or a Related Works section to introduce other works addressing the same or a similar problem. I would strongly suggest to restructure the paper entirely to give it a more structured and principled form, so that any reader can get familiar with the basics before getting into your novel contribution. Most of the parts are already there indeed, it is more a matter of giving them a coherent order of presentation to gradually lead the reader into the work.
- However, some concepts that are used throughout the paper are never discussed in practice, such as the concept of Blackwell-optimality. I feel these are important information to provide the reader with, and should find some space in a Background section.
- It is difficult to understand the analysis of results that are only reported in the Appendix. Why not moving some of the more interesting results into the main paper and discuss these, while leaving the discussion of those that do not appear there to the Appendix directly?
Typos:
- Page 5, beginning of Section 3: flashes, not fleshes

---

> ### Author Response · Authors · 2023-11-14
>
> - Regarding organization of ideas in the paper
>
> We tried a new way to convey the main ideas in a clear and concise way. There is indeed scope of improvement. As you also noted, the content is all there, some restructuring could help.
>
> In particular, we started with the plots at the beginning of the paper to remind the readers of a widely prevalent problem faced by RL practitioners that using large discount factors is problematic. Eventually we described the problem setting and the main related works, but we agree those sections can be more explicit. We will try to simplify the presentation for readers to better contextualize the problem and assess the solution.
>
> We also agree the background should contain a definition of a Blackwell-optimal policy if we use the term in the paper. At the same time, we can note that it is okay if the reader does not understand the specifics of the mathematical formulation; the implication of Blackwell’s result is more important for the paper (stated and cited at the end of page 2).
>
> - Regarding the choice of plots in the main text vs the appendix
>
> We performed numerous experiments to test our hypotheses. Given the page limits of the main text, we had to select a limited number of plots that provide a clear idea of the overall trends, and added the rest of the plots in the appendix for the more interested readers. For example, our experiments show that Centered Q-learning is robust to the choice of its parameter eta. We presented the plots for one domain in the main text (Fig 3), said the trend is repeated across the other four domains, and presented those plots in Appendix C.
>
> That being said, are there specific plots that you would suggest are more appropriate to show in the main text?
>
> Thank you again for taking the time to review this paper.

---

> > ### Comment · Reviewer_xMY5 · 2023-11-15
> > **Reply to author**
> >
> > First of all, I would like to thank the reviewer for their useful comments. Here are some additional points:
> >
> > 1. I appreciate that not all the details and useful notions can make it into the background of the paper, but the definition of Blackwell-optimality is extensively used throughout the text, and thus my suggestion to include that. Although not every mathematical detail has to be understood by the reader, this one seems to be an extremely relevant concept to your work, and thus I feel it would make sense for the reader to understand it properly.
> > 2. I appreciate the severe space constraints of the venue, but I again feel it is pointless to refer to a lot of experiments that are not actually in the paper itself. I do not have any specific experiment in mind that would have to be included in the main paper, what I am suggesting is to add a couple of additional experiments in the main body and focus your discussion on these, leaving the analysis of the others that have not been included to the appendix as well.
> >
> > I hope these additional comments help in better understanding my questions and why I feel these are important points.

---

### Official Review · Reviewer_JTzu · 2023-10-24

**Soundness:** 3 good
**Presentation:** 2 fair
**Contribution:** 2 fair
**Rating:** 5
**Confidence:** 2

**Summary:**

The authors suggest to improve Q-Learning in the continuous, infinite horizon setting, by substracting the empirical mean reward and therefore centering the reward distribution. This approach is justified by prior work, whereas the proposed method is seen as a specific parameterization of a known algorithm. This allows the authors to derive a convergence proof. Besides the theoretical results, the authors provide empirical results from three different domains.

**Strengths:**

The approach is quite general and suggest improvement potential for a wide range of RL algorithms. Additionally, the theoretical foundation improves the value of the work. The paper is well written and offers a substantial discussion of the empirical results.

**Weaknesses:**

Mostly, the weaknesses are relate to the novelty of the approach. Modifying the reward distribution is a well known approach, with either a shaping function (e.g. Learning to Utilize Shaping Rewards), substracting an empirical baseline (e.g. A2C/A3C "Asynchronous Methods for Deep Reinforcement Learning") or normalization (https://stable-baselines.readthedocs.io/en/master/guide/rl_tips.html). However, none of these methods is discussed or empirically evaluated. A theoretically comparison would also be welcome, but is not expected. Therefore, the novelty of the idea is limited and related work as well as the empirical experiments should be improved. However, the reviewer acknowledges, that the mentioned work focuses on the conventional, episodic settings and that may introduce incomparability, but this is also not discussed. Furthermore, normalization and advantage-based methods are likely applicable.

Additionally, the convergence proof is mostly based on a paper that was never peer-reviewed, as far as the reviewer can see, (Devraj, A. & Meyn, S. (2020)), weakening its reliability.

Smaller issues:
- The algorithms and domains used for the empirical evaluation are rather simple.
- Fixing epsilon without any further evaluation may lead to suboptimality and limits the fairness of the comparison.
- Considering r as part of the transition dynamics is rather unconventional. Usually, transition dynamics and reward functions are separated.
- Some references are missing (e.g. DQN)

**Questions:**

A short comment concerning advantage function and reward normalization methods would be welcome.

---

> ### Author Response · Authors · 2023-11-14
>
> - _Reward centering is compatible with advantage estimation, which solves a different problem_
>
> Reward centering and the advantage function have orthogonal benefits: the advantage function benefits the actor by reducing the variance of the updates in the policy space; reward centering benefits the critic estimation by eliminating the need to estimate the large state-independent constant offset. Note that the advantage function remains unchanged with the centered value function: $\bar{a}\_\pi^\gamma(s,a) = \bar{q}\_\pi^\gamma(s,a) - \bar{v}\_\pi^\gamma(s)$, as $\bar{q}\_\pi^\gamma(s,a) = q\_\pi^\gamma(s,a) - r(\pi)/(1-\gamma)$ and $\bar{v}\_\pi^\gamma(s) = v\_\pi^\gamma(s) - r(\pi)/(1-\gamma)$ (using the paper notation). Hence we expect the benefits of reward centering to carry over to policy-based methods. We will add this discussion to the paper.
>
> - _The idea of reward centering is also compatible with reward scaling_
>
> Scaling or centering the reward with a constant does not change the ordering of policies in a continuing problem. Scaling reduces the spread of the rewards, centering brings them close to zero, both of which can be favorable to complex function approximators such as artificial neural networks that are used for value estimation starting from a close-to-zero initialization.
> The popular stable_baselines3 repository scales (and clips) the rewards by a running estimate of the variance of the discounted returns ([line 256](https://github.com/DLR-RM/stable-baselines3/blob/master/stable_baselines3/common/vec_env/vec_normalize.py#L256)). Mean-centering the rewards as well would be beneficial for continuing domains.
>
> Note that the mechanism of tracking the mean and variance is trickier in the off-policy case than in the on-policy case. We use Wan et al.’s (2021) technique to estimate the mean, which is theoretically sound even in the off-policy case. Naively maintaining a running estimate of the variance introduces a bias (as in the stable_baselines’ approach); Schaul et al.’s (2021) technique is a good starting point. (both citations are in the paper)
>
> - _Reward centering is a kind of reward shaping_
>
> We realized this relation after submitting the paper: reward centering can be seen as reward shaping with a constant state-independent potential function: $\Phi(s) = r(\pi)/(1-\gamma) \forall s$. [Ng et al.’s (1999)](https://people.eecs.berkeley.edu/~russell/papers/icml99-shaping.pdf) Thm 1 then also says that reward centering does not change the optimal policy of the problem. A common criticism of reward shaping is that fully specifying the shaping function can be tricky, especially for large state spaces. In the case of reward centering this is relatively easy: there is a single constant value to learn and we know how to learn the average reward reliably from data. We will add this discussion to the paper.
>
> - _Limitations of the empirical study_
>
> We agree that the empirical study is limited in the number/type of domains and with variants of Q-learning. However, we expect the observed trends to hold across domains, function approximators, and with centered versions of other algorithms.
> As mention in the global comment, in this first presentation of reward centering, we tested the idea on relatively simple domains by testing performance and sensitivity-to-hyperparameters up to 50 times per experiment configuration. The resulting trends are clear across tabular, linear, and non-linear function approximations. Such systematic experiments are hard to perform on Atari-scale domains (where 3–5 runs are hence typical), but given enough compute we are confident in observing similar benefits.
>
> Reward centering can be added to any RL algorithm. We focused on Q-learning/DQN and its centered variants in this paper. We have also tried Sarsa's centered (and deep) variants and found similar improvements; a focused empirical study with centered variants of several common RL algorithms is a ripe avenue of future work.
>
> Smaller points:
> - You're right that performance will be clouded by a little exploration when epsilon is fixed with no separate evaluation period. We accepted this tradeoff in the spirit of _continual_ learning, wherein the world could be changing and hence require constant exploration and adaptation. Moreover, the comparison is fair because this choice does not favor any method in particular.
> - We apologize for the outdated arXiv reference; we will update to this peer-reviewed reference: _Devraj, A., & Meyn, S. (2021). Q-learning with uniformly bounded variance. IEEE Transactions on Automatic Control, 67(11), 5948-5963_. We will also add the missing DQN reference.
> - Considering r as part of the transition dynamics is fairly common in the RL literature, likely since Sutton & Barto (2018) started using that convention throughout their textbook (see, e.g., eqn 3.20 on page 64 of the book).
>
> Thank you for taking the time to review this paper. We hope this discussion helps answer your questions about it.

---

> > ### Comment · Reviewer_JTzu · 2023-11-16
> >
> > Thanks for the extensive comments. Due to the peer reviewed reference and the clarifications concerning the advantage function, i increased my score slightly. However, the limitations of the empirical study remain, which seems to be an issue also considered by other reviewers. Furthermore, due to the comments concerning the advantage function and the reward scaling, i deem an extension even more important to clarify in how far the proposed method still shows advantages when applied to more advanced algorithm that already resolve some issue that arise due to "difficult" reward distributions.

---

### Official Review · Reviewer_Kzst · 2023-10-24

**Soundness:** 3 good
**Presentation:** 3 good
**Contribution:** 3 good
**Rating:** 5
**Confidence:** 3

**Summary:**

The authors study the problem of rewarding centering in continuing problems. More specifically, they analyze a variant of Q-learning, *Centered Q-learning*, where the rewards experienced by the agent are subtracted from their empirical average online and incrementally.
Starting from the discounted MDP model, they develop key intuitions on why reward centering can be useful in this scenario. More specifically, starting from the Laurent series decomposition of the policy return, they highlight the presence of the average return (i.e., offset) collected by the policy. If this offset is large, approximation errors in estimating introduce problems in identifying which is the optimal action (i.e., the relative benefit of taking the optimal action). Leveraging this intuition, the authors propose the Centered Q-learning algorithm, and they analyze its convergence. Finally, they numerically validate Centered Q-learning against classical Q-learning, both in tabular cases and with linear and non-linear function approximations:
- Centered Q-Learning gains significant performance, especially in the relevant case of $\gamma \rightarrow 1$.
- Centered Q-Learning makes the algorithm robust in shifts of the reward functions
- The method is relatively robust to the choice of the update rate of the running mean of the empirical average reward

**Strengths:**

- The authors develop good intuitions on the practical values of the common practice of reward centering. These intuitions are supported by some simple theoretical analysis and experiments in several domains. Since reward centering is of practical value, I believe this work to be interesting for the community.
- The paper is well-written (with some unclear points in the theoretical part; see weakness 2 below).
- The problem is relatively novel, although some works on the topic exist.

**Weaknesses:**

1. **Empirical results** are on simple domains. I would ask the authors whether they expect their empirical results to hold in more intricated benchmarks (e.g., some Atari domain, for instance). Have the authors experimented with these more complex domains (or other domains of similar complexity)? Indeed, I retain the paper to be mainly of an empirical nature. As a consequence, I think this point could add substantial value to the submission. Currently, I believe this to be a major weakness of the work.
2. **Theoretical results** (clarity; minor). Theorem 1 shows the convergence of centered Q-learning to a particular solution. However, no comments are given on the quality of this solution. It seems to me that if, we take the arg max w.r.t. this Q-function, we obtain an optimal policy for the original discounted MDP with discount factor $\gamma$. First of all, I ask the authors if they confirm this claim. Secondly, I invite the authors to discuss it in the main paper.
3. **Theoretical results** (clarity; minor; pt. 2) Furthermore, I do not truly understand the role of $\bar{r}$ within the Theorem. As it is defined, it seems to be a free variable in a certain solution space, which makes the interpretation of Theorem 1 a bit cryptical. Could the authors discuss on this point?

**Questions:**

See weaknesses above.

---

> ### Author Response · Authors · 2023-11-14
>
> 1. _I would ask the authors whether they expect their empirical results to hold in more intricated benchmarks (e.g., some Atari domain, for instance). Have the authors experimented with these more complex domains (or other domains of similar complexity)?_
>
> We expect the observed trends to hold across domains and function approximators. The reward-centering idea is very general; the motivation (removal of the need to estimate a large state-independent offset) applies across tabular, linear, and non-linear function approximations. In this first presentation of the idea, we thoroughly tested the idea on relatively simple domains by testing performance and sensitivity-to-hyperparameters up to 50 times per experiment configuration. The resulting trends are clear across the three types of function approximations. Such systematic experiments are hard to perform on Atari-scale domains (where 3–5 runs are hence typical), but given enough compute we are confident in observing similar benefits. Continuing versions of [MinAtar](https://github.com/kenjyoung/MinAtar/) domains are a reasonable step up in complexity. We will try to report some results in the rebuttal period after first converting one or more of the appropriate episodic domains of MinAtar to continuing.
>
> 2. _It seems to me that if, we take the arg max w.r.t. this Q-function, we obtain an optimal policy for the original discounted MDP with discount factor gamma. First of all, I ask the authors if they confirm this claim. Secondly, I invite the authors to discuss it in the main paper._
>
> Indeed, reward centering does not change the ordering of the policies in continuing problems, so the argmax policy w.r.t. the centered value function is the same as argmax policy w.r.t. the original uncentered value function. In other words, reward centering does not change the final solution but makes it easy to learn by reducing the effect of the potentially large state-independent offset in the original value function. While we mention this in the paper, we can certainly make it more prominent. Thank you for pointing this out.
>
> 3. _I do not truly understand the role of $\bar{r}$ within the Theorem. As it is defined, it seems to be a free variable in a certain solution space, which makes the interpretation of Theorem 1 a bit cryptical. Could the authors discuss on this point?_
>
> $\bar{r}$ is indeed a free variable in a solution space, and for every solution in the $\bar{r}$ space, there is a corresponding solution in the value space. The ideal fixed point is the average reward of the target policy because at that point the offset is completely removed. Centered Q-learning shows one way among other possibilities in which the idea of reward centering can be combined with Q-learning. Theorem 1 says that Centered Q-learning converges to a unique tuple in the joint $\bar{r}$-values solution space. The discussion in Appendix B after the theorem proof quantifies the quality of the learned solution for the values. The quality of the corresponding $\bar{r}$ solution then follows, but we can explicitly add it to improve clarity. We hope this discussion helps with the interpretation of Theorem 1.
>
>
> Thank you for taking the time to review this paper. This discussion will help in improving the paper; we hope it helps in answering your questions as well.

---

> ### Comment · Reviewer_Kzst · 2023-11-15
> **Ack**
>
> I thank the authors for their response and the clarifications.
>
> My main concerns on the difficulty of the domains considered remain. Although these more simple experiments in which more runs are affordable are appreciated, I would also like to see whether these claims extend to more complex environments. I agree with the authors that it is reasonable to expect the results in these more intricated benchmarks to show similar trends; however, I retain that "being confident about this" is actually not enough for a paper of an empirical nature.
>
> I will take into account any eventual and additional experiments into consideration for the discussion phase. In the eventual case in which the work is rejected, I invite the authors to expand their experimental campaign for future revisions of this manuscript.
>
> I have no further questions for the authors.
>
> Best regards, \
> Reviewer Kzst

---

### Official Review · Reviewer_ecFz · 2023-11-01

**Soundness:** 3 good
**Presentation:** 3 good
**Contribution:** 3 good
**Rating:** 6
**Confidence:** 3

**Summary:**

This paper proposes a method called "Reward Centering" to improve the performance of discounted reinforcement learning (RL) algorithms. The authors argue that by centering the rewards around the current average reward, the RL algorithms can focus on the essential differences between states rather than the diffe in addition to a potentially large offset. This approach is shown to be beneficial for all discount factors, especially as the discount factor approaches 1. The paper presents empirical results on various domains, including tabular, linear, and non-linear function approximation, demonstrating the benefits of reward centering in terms of improved learning performance and robustness to shifts in rewards.

**Strengths:**

* The paper provides a clear motivation for the reward centering technique by analyzing the issues with standard RL methods when the discount factor is close to one.
* The authors present a comprehensive theoretical analysis of the convergence properties of the proposed Centered Q-learning algorithm.
* The empirical results demonstrate the benefits of reward centering across different domains and function approximation techniques, including tabular, linear, and non-linear methods.
* The paper discusses the limitations of the current approach and suggests possible extensions and improvements, such as combining reward centering with scaling techniques and extending the method to episodic problems.

**Weaknesses:**

* The paper focuses primarily on the tabular case for the theoretical analysis, and the convergence results may not directly apply to the function approximation case.
* The paper does not provide a detailed comparison of the proposed method with other state-of-the-art RL algorithms, which would help to better understand its relative performance.

**Questions:**

* How does the performance of the proposed Centered Q-learning algorithm compare to other state-of-the-art RL algorithms, such as Proximal Policy Optimization (PPO) or Soft Actor-Critic (SAC)?
* Can the reward centering technique be extended to handle episodic problems, where the ordering of policies may change due to reward shifts?
* Are there any potential drawbacks or limitations of the reward centering approach that were not discussed in the paper? For example, could centering the rewards introduce biases or affect the exploration-exploitation trade-off?
* The paper suggests that reward centering can be combined with scaling techniques to handle large magnitudes of relative values. Can you provide more details on how this combination might work and its potential benefits?

---

> ### Author Response · Authors · 2023-11-14
>
> - _How does the performance of the proposed Centered Q-learning algorithm compare to other state-of-the-art RL algorithms, such as Proximal Policy Optimization (PPO) or Soft Actor-Critic (SAC)?_
>
> Reward centering is a general technique that can be added to any RL algorithm and improve its performance. A more appropriate empirical comparison would be between PPO and Centered PPO; SAC and Centered SAC. We focused on Q-learning/DQN and its centered variants in this paper. We have also tried Sarsa and Deep Sarsa’s centered variants and found similar improvements; an extensive empirical study with centered variants of several common RL algorithms is a ripe avenue of future work!
>
> - _Can the reward centering technique be extended to handle episodic problems, where the ordering of policies may change due to reward shifts?_
>
> This is a great question and we recognize that a large fraction of the RL community would be interested in a similar simple technique for the episodic setting. We don’t have a concrete answer yet, but have identified a promising direction.
>
> - _Are there any potential drawbacks or limitations of the reward centering approach that were not discussed in the paper? For example, could centering the rewards introduce biases or affect the exploration-exploitation trade-off?_
>
> There are two potential drawbacks to reward centering. Firstly, reward centering reduces the effectiveness of optimistic initialization that is sometimes used in tabular (and some linear) implementations as an exploration technique. But for the same reason, reward centering also removes the drawbacks of inadvertent pessimistic initialization. The addition of reward centering makes RL algorithms robust to the effects of initialization—good or bad. Note that this discussion is not particularly relevant to non-linear function approximation, where generalization negates most benefits of an optimistic initialization. We allude to this effect in the paper but we can certainly make it more explicit!
>
> Secondly, reward centering introduces an additional step-size parameter ($\eta$) to any algorithm that reward centering is added to. However, our parameter studies show that performance is not particularly sensitive to the value of eta (e.g., Fig 3, or the plots in Appendix C).
>
> An attractive property of reward centering is that it does not introduce a bias because the ordering of the policies in a continuing problem is unchanged when a constant is added to all the rewards.
>
> - _The paper suggests that reward centering can be combined with scaling techniques to handle large magnitudes of relative values. Can you provide more details on how this combination might work and its potential benefits?_
>
> We haven’t worked out the exact mechanism of combining the two, but we can certainly comment on the potential benefits.
> Scaling or centering the reward with a constant does not change the ordering of policies in a continuing problem. Scaling reduces the spread of the rewards, centering brings them close to zero, both of which can be favorable to complex function approximators such as artificial neural networks that are used for value estimation starting from a close-to-zero initialization. The popular stable_baselines3 repository scales (and clips) the rewards by a running estimate of the variance of the discounted returns ([line 256](https://github.com/DLR-RM/stable-baselines3/blob/master/stable_baselines3/common/vec_env/vec_normalize.py#L256)). Mean-centering the rewards as well would be beneficial.
>
> Note that the mechanism of computing the mean and variance is trickier in the off-policy case than the on-policy case. We use [Wan et al.’s (2021)](https://arxiv.org/abs/2006.16318) technique to estimate the mean, which is theoretically sound even in the off-policy case. Naively maintaining a running estimate of the variance introduces a bias (as in the stable_baselines’ approach). [Schaul et al.’s (2021)](https://arxiv.org/abs/2105.05347) technique is a good starting point.
>
> - _The paper focuses primarily on the tabular case for the theoretical analysis, and the convergence results may not directly apply to the function approximation case._
>
> Our convergence analysis is indeed limited to the tabular case. Unfortunately, that is generally true for most RL algorithms due to our community’s limited understanding of the interaction of RL with complex function approximators, especially with the prevalent artificial neural networks.
>
> The reward-centering idea is motivated by Blackwell’s Laurent series expansion, which applies to the exact value function and not typically to approximated ones. However, our experiments with tabular, linear, and non-linear function approximation show that the implication of this theoretical result is indeed significant in the approximate empirical settings.
>
>
> Thank you for taking the time to review this paper. This discussion will help in improving the paper; we hope it helps in answering your questions as well.

---

> > ### Comment · Reviewer_ecFz · 2023-11-21
> > **Reply To The Authors**
> >
> > Dear Authors.
> >
> > Thank you for your detailed replies, the current ones solved my confusion. I will consider the comments of other reviewers and then further consider my decision.

---

### Official Review · Reviewer_DBdQ · 2023-11-01

**Soundness:** 3 good
**Presentation:** 2 fair
**Contribution:** 2 fair
**Rating:** 3
**Confidence:** 4

**Summary:**

This paper proposes a method called reward centering, which aims to solve continuing reinforcement learning problems with discounted reward criterion. The proposed method can be easily integrated with existing RL methods, although the authors have only experimented with a case of Q-learning. Reward centering involves subtracting the empirical average reward from the observations during training, which reduces the effect of a state-independent offset in the discounted value estimation. Experimental results demonstrate that reward centering can enhance performance and maintain the robustness of Q-learning in tabular cases with either linear or non-linear approximation. The paper also provides a theoretical analysis of the convergence in various types of approximation (linear and non-linear).

**Strengths:**

- It addresses a practical and essential problem of learning long-term optimal policies in continuing problems with discounting.
- It proposes a simple and effective technique that can be easily applied to existing algorithms without changing their core structure or adding much computational overhead.
- It provides a clear and rigorous theoretical analysis of the convergence and variance properties of Centered Q-learning in the tabular case.
- It presents comprehensive and convincing empirical results on various domains with different types of function approximators, showing the benefits of reward centering over standard methods.

**Weaknesses:**

- It does not extend the theoretical analysis to the function approximation case, which is more challenging and relevant for real-world applications.
- It does not compare reward centering with similar techniques that improve discounting, such as reward scaling [1], GAE [2], etc.
- The paper abuses notations: iteration number is $t$, and timestep is also $t$.
- It lacks cross-comparison from the lens of RL algorithms and hyper-parameter settings (e.g., $\epsilon$ for Q-learning).
- The motivation is not very clear. As the authors claim their proposed method solves reward shifting, but practical examples are needed to illustrate the generality of this issue.

[1] Engstrom, L., Ilyas, A., Santurkar, S., Tsipras, D., Janoos, F., Rudolph, L., & Madry, A. (2020). Implementation matters in deep policy gradients: A case study on ppo and trpo. arXiv preprint arXiv:2005.12729.

[2] Schulman, J., Moritz, P., Levine, S., Jordan, M., & Abbeel, P. (2015). High-dimensional continuous control using generalized advantage estimation. arXiv preprint arXiv:1506.02438.

**Questions:**

- Is there any difference between reward and reward rate? As the community prefers to use 'reward' to represent the environmental feedback of each decision making, so I'm not sure why the authors chose another notation if there are no differences.
- How do I choose the update frequency at lines 7 and 12 in Algorithm 3?
- I noticed the authors build Theorem 1 on top of the irreducible Markov chain, is this assumption loose or tight?
- What is $d$ in Equation 17?

---

> ### Author Response · Authors · 2023-11-14
>
> - _Reward vs reward rate?_
>
> ‘Reward rate’ denotes the rate at which the reward occurs, or the reward per step. It is used alternatively to ‘average reward’, for instance by Sutton and Barto (2018) (towards the bottom of page 249) or Wan et al. (2021), and also defined just below equation (2) in the submitted paper.
>
> - _Update frequency in lines 7,12 in Algo 3?_
>
> The weight-update and target-network-update frequency come from the DQN algorithm and not specific to the reward-centering idea. We did not tune them and set them to commonly used values in public implementations of DQN (exact values of all parameters are in Appendix C). So far, there is no exact science behind setting those deep-learning parameters.
>
> - _Irreducibility: loose or tight?_
>
> The irreducible assumption is tight, not due to theoretical necessity but for ease of presentation. It is common in non-episodic MDP literature. Our assumption builds on Devraj and Meyn's (2020), who use that assumption (Q1 in their paper). It can be relaxed to ‘weakly communicating’ at the cost of specifying convergence separately for recurrent and tranient states.
>
> - $d$ in (17)?
>
> $d$ should be $\mu$, which denotes the probability mass function defined in (12). Thank you for pointing this out.
>
> - _Extension of the analysis to function approximation_
>
> Our convergence analysis is indeed limited to the tabular case. Unfortunately, that is generally true for most RL algorithms due to our community’s limited understanding of the interaction of RL with complex function approximators, especially with neural networks.
>
> - _Comparison with scaling, GAE, etc_
>
> Reward centering is a general idea that is compatible with reward scaling and with policy-based methods; it is not a competitor. We didn’t experiment with all the combinations in this first investigation of the idea, but we can definitely make concrete predictions:
>
> (a) Scaling/clipping
>
> Scaling or centering the reward with a constant does not change the ordering of policies in a continuing problem. Scaling reduces the spread of the rewards, centering brings them close to zero, both of which can be favorable to complex function approximators such as artificial neural networks that are used for value estimation starting from a close-to-zero initialization.
> The popular stable_baselines3 repository scales (and clips) the rewards by a running estimate of the variance of the discounted returns ([line 256](https://github.com/DLR-RM/stable-baselines3/blob/master/stable_baselines3/common/vec_env/vec_normalize.py#L256)). Mean-centering the rewards as well would be beneficial for continuing domains. That being said, more work needs to be done to soundly estimate the variance in the off-policy continuing case (stable_baselines’ running estimate introduces a bias). [please see additional discussion about scaling with reviewer ecFz; on shaping with JTzu]
>
> Reward clipping in general changes the problem. Blinding the agent from large rewards can impose a performance ceiling or make problems impossible to solve (see Schaul et al.'s (2021) Sec 4.3 discussion on Atari).
>
> (b) Advantage Estimation
>
> Reward centering and the advantage function have orthogonal benefits: the advantage function benefits the actor by reducing the variance of the updates in the policy space; reward centering benefits the critic estimation by eliminating the need to estimate the large state-independent constant offset. Note that the advantage function remains unchanged with the centered value function: $\bar{a}\_\pi^\gamma(s,a) = \bar{q}\_\pi^\gamma(s,a) - \bar{v}\_\pi^\gamma(s)$, as $\bar{q}\_\pi^\gamma(s,a) = q\_\pi^\gamma(s,a) - r(\pi)/(1-\gamma)$ and $\bar{v}\_\pi^\gamma(s) = v\_\pi^\gamma(s) - r(\pi)/(1-\gamma)$ (using the paper notation). Hence we expect the benefits of reward centering to carry over to policy-based methods.
>
> We will add this discussion to the paper.
>
> - _...lacks cross-comparison..._
>
> Could you please explain what you mean by cross comparison in this context?
>
> - _...the proposed method solves reward shifting, but practical examples are needed to illustrate the generality of this issue._
>
> Shifting the rewards of a continuing problem does not change the problem (the ordering of policies remains the same). However, our experiments with multiple domains reveal that common algorithms behave significantly differently when the rewards are shifted. This is undesirable. If the problem doesn’t change, the performance of solution methods shouldn’t change either. So while an RL practitioner doesn’t typically shift the rewards, it is a hidden hyperparameter that affects the speed of learning. We are happy to note the reward-centering idea to solve a different problem of large gamma-dependent offsets also makes methods robust to any shifts in the rewards.
>
> Thank you for taking the time to review this paper. This discussion will help in improving the paper; we hope it helps in answering your questions as well. (all the references are from the paper)

---

> > ### Comment · Reviewer_DBdQ · 2023-11-21
> > **Thanks for your response**
> >
> > I thank the authors for their clarification, which resolves some of my doubts, and I admit this work may be an interesting finding relating to the training efficiency of RL. The cross-comparison I mentioned in my previous comments indicates a comparison of different hyper-parameter choices. Indeed, this is not a mandatory requirement, but it would be nice to have. As for the comparison with existing reward scaling methods, I still look forward to seeing the corresponding experiments, as a confirmation of your predictions.

---

### Author Response · Authors · 2023-11-14

Reward centering is a general idea. It can be added to any standard RL algorithm for solving continuing problems. The theory suggests its addition makes the learning problem easier without changing the solution. We demonstrated the benefits of reward centering in tabular, linear, and non-linear experiments with variants of the Q-learning algorithm. Given the generality of the idea, we expect similar trends to hold with centered versions of other algorithms and across various domains.

Based on the reviewers’ comments, we have decided to make the following changes to improve the quality of the paper. We will soon upload a new PDF with these changes.
- Addition of a discussion of how the general idea of reward centering is compatible with (and not a competitor of) approaches like scaling and advantage estimation.
- Clarification of Theorem 1: on the tightness of the irreducible assumption and what it means for the algorithm to converge to one of multiple solutions.
- Adding a discussion of an implication of reward centering: how it reduces the effect of initialization—positive or negative.

We discuss these points in detail in replies to individual reviewer comments, where we also discuss minor fixes in notations and citations to improve readability.

We thank all the reviewers for their time in helping improve this paper. We hope the rest of the discussion helps in improving everyone’s understanding of the general idea of reward centering.

---

> ### Author Response · Authors · 2023-11-23
> **PDF updated**
>
> We have updated the PDF to address two main concerns and a few small clarifications:
> 1. We've highlighted that the focus of the paper is not Centered Q-learning but the general reward-centering idea that can be applied to any continuing RL algorithm that estimates values. To this effect, we've added a line to the abstract, a paragraph at the end of Section 2, and a couple of sentences in the concluding section.
> 2. We've added a discussion on related approaches in the appendix: how reward centering is compatible with (and not a competitor of) reward scaling and advantage estimation, and how it is a form of state-independent shaping.
>
> The smaller clarifications include a discussion of the irreducibility assumption and a note on the quality of the reward-rate estimate. We also updated some references and fixed a couple of typos.
>
> We thank the reviewers for their time in going through our responses. We hope the discussion (and a final look at the updated PDF) has given a clearer idea of the appeal and the benefits of reward centering.

---

### Meta-Review · Area_Chair_5D1Y · 2023-12-06

**Metareview:**

The paper proposes a novel technique of reward centering to improve the performance of RL methods with discounting. Theoretical analysis is provided for the tabular Q-learning method with reward centering. The reviewers acknowledged that the proposed technique of reward centering is of practical interest and can be easily applied to existing RL algorithms. However, the reviewers pointed out several weaknesses in the paper, and raised concerns related to the limited scope of theoretical results and limited empirical evaluation. We want to thank the authors for their detailed responses. Based on the raised concerns and follow-up discussions, unfortunately, the final decision is a rejection. Nevertheless, the reviewers have provided detailed and constructive feedback. We hope the authors can incorporate this feedback when preparing future revisions of the paper.

**Justification For Why Not Higher Score:**

The reviewers pointed out several weaknesses in the paper, and raised concerns related to the limited scope of theoretical results and limited empirical evaluation. A majority of the reviewers think that the work is not yet ready for publication.

**Justification For Why Not Lower Score:**

N/A

---

### Decision · Program_Chairs · 2024-01-16

Reject